# Risk Factors for Mortality in Cardiac Implantable Electronic Device (CIED) Infections: A Systematic Review and Meta-Analysis

**DOI:** 10.3390/jcm11113063

**Published:** 2022-05-29

**Authors:** Jinghao Nicholas Ngiam, Tze Sian Liong, Meng Ying Sim, Nicholas W. S. Chew, Ching-Hui Sia, Siew Pang Chan, Toon Wei Lim, Tiong-Cheng Yeo, Paul Anantharajah Tambyah, Poay Huan Loh, Kian Keong Poh, William K. F. Kong

**Affiliations:** 1Division of Infectious Diseases, Department of Medicine, National University Health System, Singapore 119228, Singapore; nicknjh1311@hotmail.com (J.N.N.); mdcpat@nus.edu.sg (P.A.T.); 2Department of Medicine, National University Health System, Singapore 119228, Singapore; liongtzesian@gmail.com (T.S.L.); meng_ying_sim@nuhs.edu.sg (M.Y.S.); 3Department of Cardiology, National University Heart Centre Singapore, National University Health System, Singapore 119074, Singapore; nicholas_ws_chew@nuhs.edu.sg (N.W.S.C.); ching_hui_sia@nuhs.edu.sg (C.-H.S.); toon_wei_lim@nuhs.edu.sg (T.W.L.); tiong_cheng_yeo@nuhs.edu.sg (T.-C.Y.); poay_huan_loh@nuhs.edu.sg (P.H.L.); kian_keong_poh@nuhs.edu.sg (K.K.P.); 4Yong Loo Lin School of Medicine, National University of Singapore, Singapore 117597, Singapore; mdccsp@nus.edu.sg; 5Cardiovascular Research Institute, National University Health System, Singapore 119074, Singapore; 6Infectious Diseases Translational Research Programme, Department of Medicine, Yong Loo Lin School of Medicine, National University of Singapore, Singapore 117597, Singapore

**Keywords:** cardiac implantable electronic device, infection, mortality, outcomes

## Abstract

Background: Infections following cardiac implantable electronic device (CIED) implantation can require surgical device removal and often results in significant cost, morbidity, and potentially mortality. We aimed to systemically review the literature and identify risk factors associated with mortality following CIED infection. Methods: Electronic searches (up to June 2021) were performed on PubMed and Scopus. Twelve studies (10 retrospective, 2 prospective cohort studies) were included for analysis. Meta-analysis was conducted with the restricted maximum likelihood method, with mortality as the outcome. The overall mortality was 13.7% (438/1398) following CIED infection. Results: On meta-analysis, the male sex (OR 0.77, 95%CI 0.57–1.01, I^2^ = 2.2%) appeared to have lower odds for mortality, while diabetes mellitus appeared to be associated with higher mortality (OR 1.47, 95%CI 0.67–3.26, I^2^ = 81.4%), although these trends did not reach statistical significance. Staphylococcus aureus as the causative organism (OR 2.71, 95%CI 1.76–4.19, I^2^ = 0.0%), presence of heart failure (OR 1.92, 95%CI 1.42–4.19, I^2^ = 0.0%) and embolic phenomena (OR 4.00, 95%CI 1.67–9.56, I^2^ = 69.8%) were associated with higher mortality. Surgical removal of CIED was associated with lower mortality compared with conservative management with antibiotics alone (OR 0.22, 95%CI 0.09–0.50, I^2^ = 62.8%). Conclusion: We identified important risk factors associated with mortality in CIED infections, including *Staphyloccocus aureus* as the causative organism, and the presence of complications, such as heart failure and embolic phenomena. Surgery, where possible, was associated with better outcomes.

## 1. Introduction

There has been an increase in the incidence of cardiac implanted electronic devices (CIED) being used in a variety of cardiac disorders [1]. These would include anti-bradycardic CIED (pacemaker), as well as implantable cardioverter-defibrillators (ICD) [2]. In tandem, infections associated with these devices have also shown an increasing trend over the years, with significant mortality risk and morbidity [3,4]. The true incidence of CIED infections is not well studied but has been reported to be as high as 1.9 per 1000 device-years, in a large population-based study of 1524 subjects [5].

The standard of care for patients with CIED infections would be to explant the infected device for adequate source control where possible and administer systemic antibiotics [6]. prior to reimplantation. CIED infections remain significantly morbid, with high associated healthcare costs and a reported risk of in-hospital mortality of up to 11.3% [7,8]. Several studies have identified risk factors for the development of CIED infections, for example, older age, end-stage kidney disease, and poorly controlled diabetes mellitus [9]. In addition, peri-implant complications, such as pocket hematoma requiring repeat intervention have also been associated with higher rates of device infection [10,11].

Several strategies have been employed to reduce this infection risk, including the use of prophylactic antibiotics [12]. In a meta-analysis of five studies, the use of antibiotic envelopes helped to significantly reduce the risk of CIED infections (relative risk reduction of 69%) [13]. Most centres also employ strict infection control practices to reduce the incidence of CIED infections, though the effect of such measures remains to be rigorously studied [14].

Several studies have attempted to identify risk factors associated with mortality in patients with CIED infection. Older age, renal failure, heart failure, steroid use and the presence of infective endocarditis have been associated with mortality [15]. However, the majority of the studies are of small sample sizes and are limited by the retrospective nature of their study design. They are also often single-centre studies that examine only a few variables [16,17]. To our knowledge, there has not been a prior meta-analysis on the mortality risk factors in patients with CIED infections. Therefore, we performed a meta-analysis based on available evidence on risk factors associated with mortality in patients diagnosed and treated for CIED infections.

## 2. Methods

### 2.1. Data Sources

Two reviewers (JNN and WKFK) independently performed a systematic literature search on PubMed and Scopus, for manuscripts published up to June 2021. The following search terms were applied (pacemaker * OR defibrillator *) AND (infect * OR endocarditis) AND (outcome * OR mortality *). The reference list of each relevant article was also subsequently searched manually. We examined articles published in the English language and did not include unpublished studies presented as conference abstracts.

### 2.2. Study Selection

The following inclusion criteria were used to identify studies that would be included for review. We examined risk factors based on the univariate analysis of each study. The studies had to (i) examine potential risk factors for mortality in patients with CIED infection; (ii) with a CIED defined as either an implanted anti-bradycardic CIED (pacemaker) or defibrillator. (iii) The studies could be either retrospective or prospective, (iv) and we only considered overlapping studies if they had examined distinct risk factors. (v) Studies on paediatric patients were excluded. (vi) Patients treated medically or surgically for CIED infection were both considered eligible. We excluded studies on cardiac resynchronisation therapy device infections, as these patients often had advanced heart failure and formed a distinct population with a significantly higher risk of infection and consequently mortality as well [18].

### 2.3. Data Extraction

The two reviewers (JNN and WKFK) independently extracted the following data: design and year of study, population characteristics, risk factors for mortality including gender, device type, diabetes mellitus, positive blood culture, organism cultured, presence of embolic phenomena, heart failure, and surgical procedure. We resolved any disagreement with a consensus meeting by the authors.

### 2.4. Definition

We adopted the proposed Mayo CIED infection criteria and classification [19]. CIED infection was defined as the (i) presence of local signs of inflammation at the generator pocket, or (ii) CIED-related endocarditis. Endocarditis was clinically confirmed by the presence of valvular or lead vegetations in echocardiography, by meeting the modified Duke criteria [20]. This was also coherent with the European Heart Rhythm Association (EHRA) international consensus document on how to prevent, diagnose and treat CIED infections [21].

### 2.5. Statistical Methods

The meta-analyses [22], estimated with the restricted maximum likelihood (REML) method, were carried out with the occurrence of death as the outcome. The synthesised odds ratios (OR) were computed from the comparisons of (a) diabetes versus no diabetes, (b) male versus female, (c) implantable cardioverter defibrillator versus anti-bradycardic CIED (pacemaker), (d) *Staphylococcus aureus* versus non-*Staphylococcus aureus* infection, (e) embolism (either left- or right-sided embolic phenomena) versus no embolism, (f) heart failure versus no heart failure, (g) surgery versus conservative management with antibiotic therapy alone. For clarity of graphical representation, the ORs in the figures were presented as log (OR). The cut-off for significance was, therefore, 0 for the log (OR) presented. The default random-effect method was applied, in view of the multiple sources of differences in study designs (i.e., prospective and retrospective), patient populations, and analytical methods reported in the papers considered for analyses. However, the chi-square tests, Tau^2^, I^2^ and H^2^ measures were reported routinely for examining the heterogeneity issue. The results were summarised with forest plots, with funnel plots generated to give some indication of publication bias. Data were analysed with Stata MP version 17 (Stata Corporation, College Station, TX, USA), all statistical analyses were conducted at a 5% level of significance or its equivalence with 95% confidence intervals.

## 3. Results

A database search on PubMed and Scopus yielded 1426 possible results, of which 513 were screened, and subsequently, 12 studies met the inclusion criteria and were included in our meta-analysis (Figure 1). Of the 12 studies included, two were prospective cohort studies, while the remaining 10 were retrospective cohort studies (Table 1). There were no randomised controlled trials. The overall mortality was 13.7% (438/1398) following CIED infection.

Patients who were infected with *Staphylococcus aureus* faced a significantly higher odds of death, compared with those who were infected with other organisms (OR: 2.714, 95% C.I.: 1.759–4.187) (Figure 2). All five studies (one prospective and four retrospective) report a similar finding concerning the risk factor (I^2^ = 0.0%). There is no evidence of publication bias (Appendix A).

Based on four studies (one prospective and three retrospective), the next meta-analysis yielded a significantly higher odds of death for patients suffering from embolism (OR: 4.000, 95% C.I.: 1.673–9.564) (Figure 3). There is evidence of heterogeneity (I^2^ = 69.8%) and publication bias (Appendix A). Care must be taken in interpreting the findings as the analysis involves few reported results.

Heart failure was also found to be a significant risk factor of death, with the odds nearly doubled (OR: 1.922, 95% C.I.: 1.419–2.603) (Figure 4). The meta-analysis could suffer from publication bias (Appendix A), although the evidence is relatively weak.

Compared with their female counterparts, male subjects had a similar rate of death (OR: 0.771, 95% C.I.: 0.586–1.014) (Figure 5). Although this did not achieve statistical significance, there appeared to be a trend toward the female gender being associated with a higher risk of mortality. As such, this may be an important consideration for a future prospective study. There was no strong evidence of publication bias (I^2^ = 2.2%) and heterogeneity (Appendix A).

The meta-analysis comparing patients with or without diabetes mellitus yielded a combined OR of 1.474 (95% C.I.: 0.665–3.264). A total of six studies (one prospective and five retrospective) were considered, and there is evidence of heterogeneity (I^2^ = 81.4%) and publication bias (Appendix A). This suggests that diabetes is not a significant risk factor for death in patients with CIED infections, although diabetes has been shown in a number of studies to be a risk factor for infection (Figure 6, Appendix A) [21].

Subjects with ICD infections had a similar risk of death when compared with those with infected anti-bradycardic CIED (pacemakers) (OR: 0.672, 95% C.I.: 0.338–1.334). (Figure 7). There is evidence of heterogeneity (I^2^ = 75.6%) and publication bias (Appendix A) in this meta-analysis involving six studies (one prospective and five retrospective).

Based on the results from 11 studies (two prospective and 11 retrospective), the synthesised odds ratio of 0.218 (95% C.I.: 0.091–0.503) (Figure 8) suggests that patients who had surgical management of their CIED infections fared better than those treated medically although there is the risk of selection bias. There is evidence of publication bias (Appendix A) and heterogeneity (I^2^ = 62.8%).

## 4. Discussion

While several studies have begun to define the risk factors for the development of CIED infection in patients with cardiac devices, to our knowledge, our study is the first to examine the risk factors for mortality in the population who have already developed CIED infection. This was a clinically important disease entity, where mortality may be as high as 5.6–11.3% [15]. We examined several patient factors, device factors and complications that may be associated with mortality in patients with CIED infections.

Although diabetes is a known risk factor for infection including CIED infection, it was not associated with mortality [32]. While males and females with CIED infections had similar mortality rates, it has been widely recognised that cardiac disease is often under-recognised in females. This may lead to females presenting later and with more advanced disease, consequently with poorer outcomes [33,34].

There was no difference between PPM and ICD infections in terms of mortality rates. This might be because the most serious infections are related to endocarditis which occurs through the seeding of bacteria through the leads on the tricuspid valve or the wall of the right ventricle or atrium. As both types of devices use similar leads, although some are dual lead, it may be that the impact of mortality is more due to the number of leads rather than the device per se. There was insufficient information provided in the articles to determine the role of dual lead vs single lead devices in the severity of the infection and consequent mortality. At current, the studies are not able to adequately discriminate this and it remains a topic for future study.

In terms of microbiology, it was not surprising that we found that patients when CIED infection was caused by *Staphylococcus aureus*, it was significantly and consistently associated with a higher risk of mortality (OR 2.71, 95%CI 1.76–4.19). It has been well established that *Staphylococcus aureus* endocarditis has a high rate of mortality of up to 20–30% [35,36]. Several factors likely contribute to the high mortality rate, including the organism’s virulence, rising rates of antimicrobial resistance, and the organism’s propensity to adhere to prosthetic material in patients [37]. Special vigilance should, therefore, be given to patients with *Staphyloccocus aureus* CIED infections. Further study would be important to compare outcomes between methicillin-susceptible (MSSA) and methicillin-resistant *Staphylococcus aureus* (MRSA) infections and to compare the different treatment regimens including vancomycin versus anti-*Staphylococcal* penicillins or the adjunctive use of gentamicin and rifampicin.

Complications observed following CIED infection, such as heart failure (OR 1.92, 95%CI 1.42–2.60) and embolic phenomena (OR 4.00, 95%CI 1.67–9.56) have also been consistently associated with higher mortality in our meta-analysis. The presence of heart failure and embolism (either left- or right-sided embolic phenomena) may suggest the presence of infective endocarditis or large lead vegetations that indicate more severe CIED infection [38]. In most guidelines for the management of patients with infective endocarditis, the presence of heart failure is the major indication for surgery to prevent in-hospital mortality [39].

Finally, in terms of management of CIED infection, surgical removal remains the cornerstone in addition to effective antibiotics [40]. Indeed, in the studies examined, we consistently found that patients who had undergone surgery were at a significantly lower risk of mortality (OR 0.22, 95%CI 0.09–0.50). However, this observation may in part be explained by the fact that patients who were frail and deemed to be poor surgical candidates were more likely to pass away from CIED infection. Nevertheless, in patients in whom it is possible, our findings support the notion that surgical removal of the infected CIED would be recommended in the management of CIED infections [41].

## 5. Limitations

A number of limitations should still be taken into consideration. Although we systematically analysed the available literature on risk factors for mortality in CIED infections, the number of publications remains few. Most of the cohorts studied were also only small to moderately sized. Furthermore, there may be publication bias, particularly if data on some risk factors remain unpublished due to a lack of a significant trend identified. We routinely checked and reported the presence of publication bias and the presence of heterogeneity. All the studies included were observational, with the majority being retrospective in nature. The identified risk factors associated with mortality were, therefore, no direct evidence of causality and caution in interpretation would be needed. There had not been enough studies to discriminate or compare pocket infection with endocarditis in the context of CIED infections. We had also deliberately excluded studies on cardiac resynchronisation therapy device infections, as these patients had universally and advanced underlying cardiac disease. Infections were significantly more common with these devices, and the advanced cardiac disease further meant that these infections had higher mortality than infections with other cardiac devices [18]. Furthermore, due to the underlying significant comorbidity, these devices were less likely to be explanted. Future prospective studies are warranted to characterise this distinct population and explore other associations.

## 6. Conclusions

We performed a meta-analysis of the available literature identifying risk factors associated with mortality in CIED infections. The presence of *Staphyloccocus aureus* CIED infection, and complications, such as heart failure and embolic phenomena were consistently associated with mortality. Surgical management, where possible, was associated with better outcomes. Strategies to optimize the treatment and prevention of CIED infections are needed to reduce the significant mortality rate associated with these important life saving devices.

## Figures and Tables

**Figure 1 jcm-11-03063-f001:**
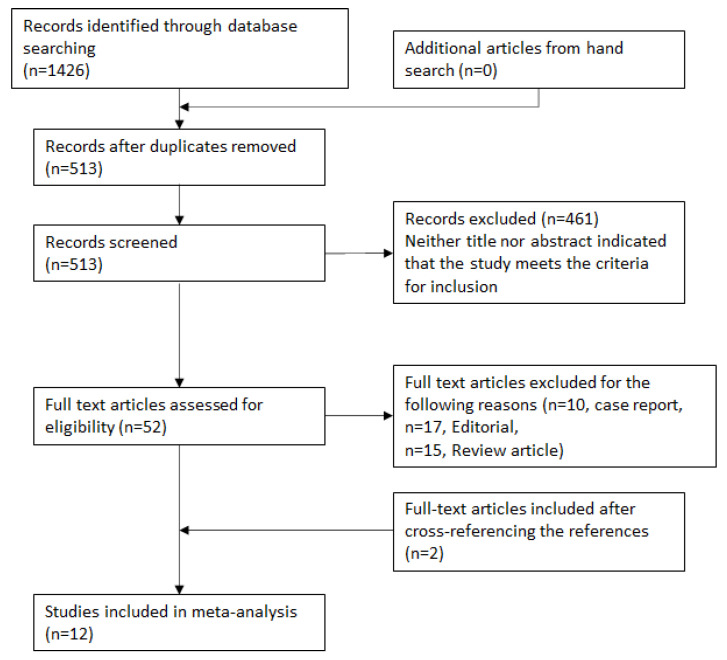
Flowchart outlining study selection process.

**Figure 2 jcm-11-03063-f002:**
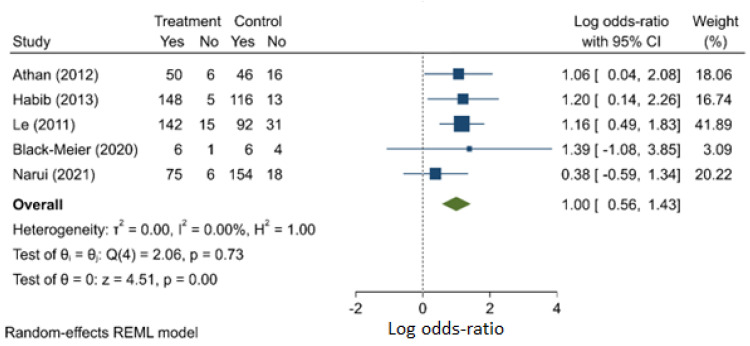
*Staphylococcus aureus* compared with non-*Staphylococcus aureus* CIED infection as a risk factor for mortality.

**Figure 3 jcm-11-03063-f003:**
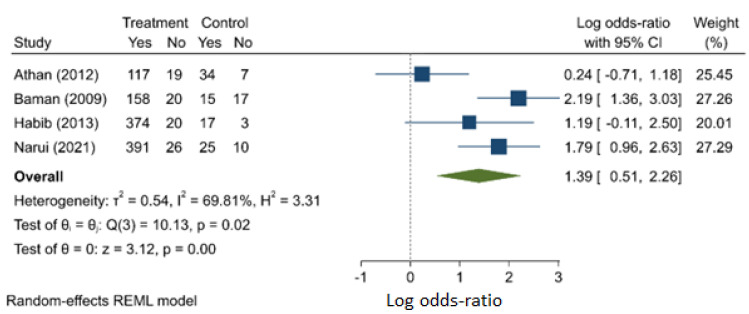
Presence of embolism/embolic phenomena as a risk factor for mortality.

**Figure 4 jcm-11-03063-f004:**
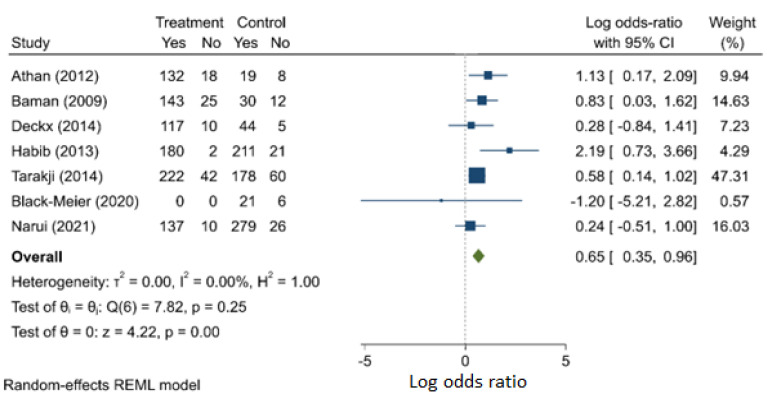
Heart failure as a risk factor for mortality.

**Figure 5 jcm-11-03063-f005:**
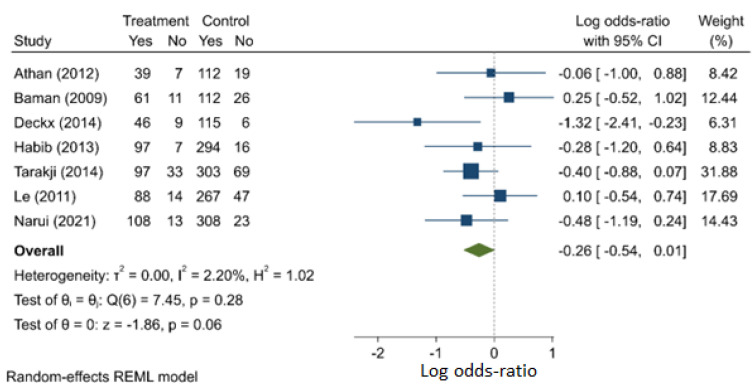
Males compared with females as a risk factor for mortality.

**Figure 6 jcm-11-03063-f006:**
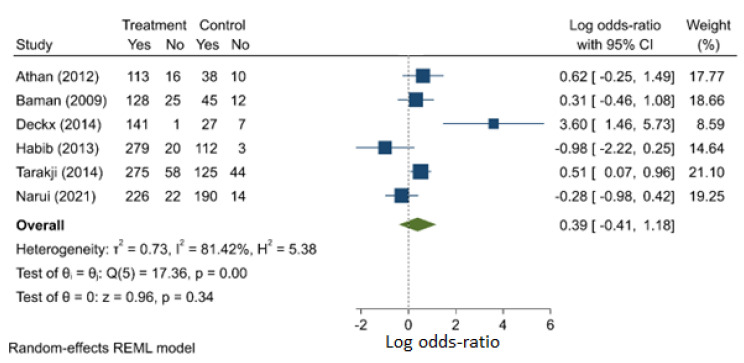
Presence of diabetes mellitus as risk factor for mortality.

**Figure 7 jcm-11-03063-f007:**
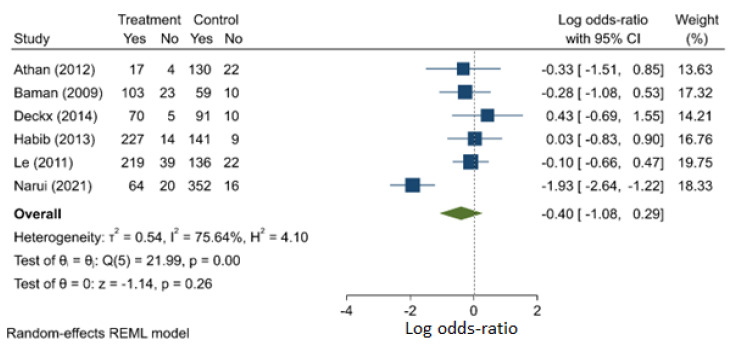
Implantable cardioverter defibrillator compared with anti-bradycardic CIED (pacemaker) as a risk factor for mortality.

**Figure 8 jcm-11-03063-f008:**
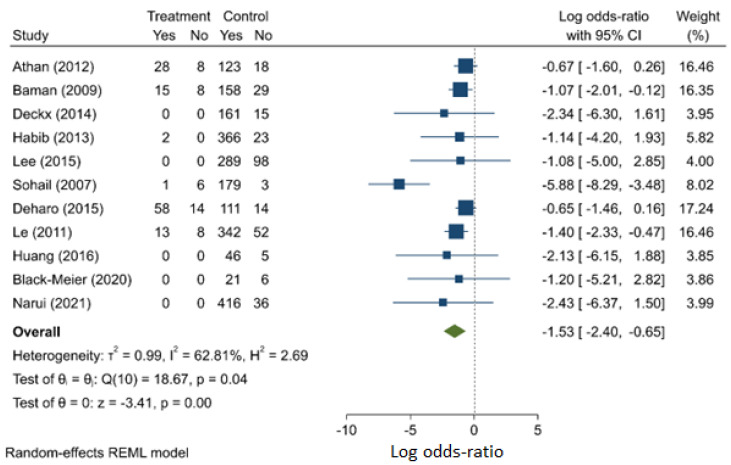
Surgery compared with medical therapy as risk factor for mortality.

**Table 1 jcm-11-03063-t001:** Characteristics of the included studies.

Study/Year	Study Design	Sample Size	Risk Factors Identified
Sohail 2007 [8]	Retrospective	387	*Staphyloccocus aureus*, surgery *
Baman 2009 [23]	Retrospective	210	Diabetes, device type, gender, embolism, heart failure, surgery
Le 2011 [24]	Retrospective	416	Device type, gender, *Staphylococcus aureus*, surgery *
Athan 2012 [25]	Prospective	177	Diabetes, device type, gender, *Staphylococcus aureus*, embolism, heart failure, surgery *
Habib 2013 [15]	Retrospective	414	Diabetes, device type, gender, *Staphylococcus aureus*, embolism, heart failure, surgery *
Deckx 2014 [26]	Retrospective	176	Diabetes, device type, gender, heart failure, surgery *
Tarakji 2014 [27]	Retrospective	502	Diabetes, gender, heart failure
Lee 2015 [28]	Retrospective	387	*Staphylococcus aureus*, surgery *
Deharo 2015 [29]	Prospective	197	Diabetes, gender, device type, surgery *
Huang 2016 [30]	Retrospective	51	Surgery *
Black-Maier 2020 [31]	Retrospective	27	Diabetes, gender, *Staphylococcus aureus*, surgery *
Narui 2021 [16]	Retrospective	452	Diabetes, gender, device type, *Staphylococcus aureus*, embolism, heart failure, surgery *

* Surgical intervention identified as a protective factor for mortality.

## Data Availability

Data may be made available on request from the corresponding author.

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
