# Peer review of "Risk Factors for Mortality in Cardiac Implantable Electronic Device (CIED) Infections: A Systematic Review and Meta-Analysis"

_jcm, 2022, doi:10.3390/jcm11113063_

Round 1

Reviewer 1 Report

CIED infections are of major importance for GP practitioners, internal medicine specialists and cardiologists.

My major remark is the lack of clear chart flow of the included studies. It is written that 513 possible results were identified, of which 12 studies were included. I have just copied and paced used formula to PubMed and I identified 1980 potential manuscripts. I doubt that a number of manuscripts doubled within less than one year. Moreover, I easily found one trial which, I think, fulfill the inclusion criteria, but is not included in the meta-analysis: https://pubmed.ncbi.nlm.nih.gov/30091136/   Therefore, I doubt whether the meta-analysis was conducted in the proper way. 

Author Response

Manuscript ID: jcm-1699583

Title: Risk factors for mortality in cardiac implantable electronic device (CIED) infections: A systematic review and meta-analysis

Journal of Clinical Medicine

We thank the Editor for allowing us the opportunity to revise our manuscript and the Reviewers for the important and constructive comments. We have amended our paper in order to address the points raised by the Reviewers.

In the sections below, each of the points raised is identified and addressed with changes in the revised manuscript.

Reviewer #1:

CIED infections are of major importance for GP practitioners, internal medicine specialists and cardiologists.

My major remark is the lack of clear chart flow of the included studies. It is written that 513 possible results were identified, of which 12 studies were included. I have just copied and paced used formula to PubMed and I identified 1980 potential manuscripts. I doubt that a number of manuscripts doubled within less than one year. Moreover, I easily found one trial which, I think, fulfill the inclusion criteria, but is not included in the meta-analysis: https://pubmed.ncbi.nlm.nih.gov/30091136/   Therefore, I doubt whether the meta-analysis was conducted in the proper way. 

We thank the Reviewer for this important comment. Indeed, we agree that a major limitation is the small number of relevant studies. We have since included a flowchart to demonstrate the study selection process (Figure 1).

As for the manuscript/trial raised by the Reviewer, we wish to clarify that during the study selection process, we had excluded studies that purely examined CRT devices. These patients often had advanced heart failure, and was in itself associated with a significantly higher infection risk, and consequently also a higher mortality risk.1 This risk was often orders of magnitude higher than with other cardiac devices (e.g. pacemakers and implantable cardiac defibrillators). In the study mentioned by the Reviewer, the was mortality 51 – 75%, compared with the pooled mortality of 13% that we had observed in the studies we reviewed. The patients with CRT infections were also significantly less likely to have their devices explanted given their underlying significant comorbidity. We had instead aimed to compare ICDs with pacemakers (anti-bradycardic CIEDs). Infections in patients with CRT devices likely form a distinct population which would be important to study in detail, but had not been the focus of our current meta-analysis.

We have added these points to our methods for greater clarity, as well as added this to our limitations.

1 Olsen T, Jørgensen OD, Nielsen JC, Thøgersen AM, Philbert BT, Johansen JB. Incidence of device-related infection in 97 750 patients: clinical data from the complete Danish device-cohort (1982-2018). Eur Heart J. 2019;40(23):1862-1869.

We thank the Reviewer and Editor for the kind and helpful comments. We hope the paper is now suitable for publication in the Journal.

Thank you

Reviewer 2 Report

The manuscript by Ngiam et al. represents a systematic review and meta-analysis of the available studies examining the risk factors for mortality in patients with CIED infections. Out of 513 possible publications they could identify 12 studies which corresponded to the prespecified criteria. These studies were included in the meta analysis which used the default random-effects model with death as the outcome. The authors compared the patients with regards to the presence of several clinical characteristics. These were the presence of diabetes mellitus, heart failure, S. Aureus infection, gender, embolic events, type of device and the need for surgical treatment. Results demonstrate the higher mortality risk in patients with S. Aureus infection, heart failure. Female gender was also associated with increased mortality. Although this meta-analysis adds to the existing evidence on factors determining mortality risk in patients with CIED infection the results should be interpreted with great caution due to the relatively small cumulative number of patients coming from observational, mainly retrospective studies. There is also publication bias as reported by the authors. All these limitations have been recognized by the authors which call for further prospective studies in the field. Generally the manuscript is easy to read although the results part need major reworking in order to bring more clarity to the reader. Specific comments are outlined below:

  1. The paragraph on definition of CIED in the Methods section should provide reference to the most recent criteria and use them for defining CIED infection (Blomstrom-Lundqvist et al. doi:10.1093/europace/euz246).
  2. Page 3, line 132-134. The sentence is unclear to the reader and should be rephrased.
  3. Lines 142-143. Please replace “permanent pacemaker” with “antibradycardic CIED” or similar term.
  4. Table 1. The whole table should be reworked, mainly in terms of formatting. There should be references to the included studies. Currently these cannot be found in the reference list.
  5. All the figures should be reworked by placing a vertical line at the value of 1 to better signify this value. Using log-scale and reporting linear ORs in the text brings confusion and should be changed.

6. References should be numbered in Arabic numerals and all the references should follow the recommended style as pointed out in the author instructions.

Author Response

Manuscript ID: jcm-1699583

Title: Risk factors for mortality in cardiac implantable electronic device (CIED) infections: A systematic review and meta-analysis

Journal of Clinical Medicine

We thank the Editor for allowing us the opportunity to revise our manuscript and the Reviewers for the important and constructive comments. We have amended our paper in order to address the points raised by the Reviewers.

In the sections below, each of the points raised is identified and addressed with changes in the revised manuscript.

Reviewer #2:

The manuscript by Ngiam et al. represents a systematic review and meta-analysis of the available studies examining the risk factors for mortality in patients with CIED infections. Out of 513 possible publications they could identify 12 studies which corresponded to the prespecified criteria. These studies were included in the meta analysis which used the default random-effects model with death as the outcome. The authors compared the patients with regards to the presence of several clinical characteristics. These were the presence of diabetes mellitus, heart failure, S. Aureus infection, gender, embolic events, type of device and the need for surgical treatment. Results demonstrate the higher mortality risk in patients with S. Aureus infection, heart failure. Female gender was also associated with increased mortality. Although this meta-analysis adds to the existing evidence on factors determining mortality risk in patients with CIED infection the results should be interpreted with great caution due to the relatively small cumulative number of patients coming from observational, mainly retrospective studies. There is also publication bias as reported by the authors. All these limitations have been recognized by the authors which call for further prospective studies in the field. Generally the manuscript is easy to read although the results part need major reworking in order to bring more clarity to the reader. Specific comments are outlined below:

The paragraph on definition of CIED in the Methods section should provide reference to the most recent criteria and use them for defining CIED infection (Blomstrom-Lundqvist et al. doi:10.1093/europace/euz246).

We thank the reviewer for this important comment. The EHRA international consensus document in CIED infections coheres with the Mayo definition which we had adopted for our study. We have included this definition in our Methods.

Page 3, line 132-134. The sentence is unclear to the reader and should be rephrased.

We thank the reviewer for this important comment. The sentence has been rephrased for clarity.

Lines 142-143. Please replace “permanent pacemaker” with “antibradycardic CIED” or similar term.

We thank the Reviewer for this helpful suggestion. We have replaced the term with pacemaker (anti-bradycardic CIED) where appropriate, for greater understanding and clarity.

Table 1. The whole table should be reworked, mainly in terms of formatting. There should be references to the included studies. Currently these cannot be found in the reference list.

We have reworked this table to include a list for references where the included studies could be found.

All the figures should be reworked by placing a vertical line at the value of 1 to better signify this value. Using log-scale and reporting linear ORs in the text brings confusion and should be changed.

The Reviewer indeed raises an important point. We had to use a log scale as given our relatively small sample size and few studies, some of the ORs were significantly wide. Using a linear scale would instead have been even more confusing and distracting, given the wide confidence intervals. Indeed, a vertical line is visually very helpful for the figures, and we have since placed a vertical line at the value of 0 for greater clarity (log (1) = 0) for each of the figures.

  1. References should be numbered in Arabic numerals and all the references should follow the recommended style as pointed out in the author instructions.

We thank the Reviewer for this important feedback. The reference list has since been updated accordingly and labelled using Arabic numerals. It was indeed labelled with Arabic numerals in the original submission, but may have been inadvertently switched to Roman numerals during the copy-editing process.

We thank the Reviewer and Editor for the kind and helpful comments. We hope the paper is now suitable for publication in the Journal.

Thank you

Round 2

Reviewer 2 Report

Following a major revision the manuscript of Ngiam et al. has improved greatly. The authors have responded to the issues raised by the reviewer and have made the necessary changes in the text. However there are still some points that need to be considered.

1. The references to the studies in the table should be numbered in Arabic numbers and appear as cited in the reference list. Listing them at the bottom of the table is redundant and just makes it harder to follow the text.

2. I do understand the value of logOR in this case but the authors should specify that a log OR has been used in the legends. I still think that this way of representing brings some unclarity and confusion as in the abstract and in the text the ORs are reported as linear measures. Perhaps this should be specified somehow in the text as well – e.g. by pointing that specifically in the statistical analysis section. It should also be specifically pointed out that the cut-off for LogOR is 0 as the reader gets confused very easily when interpreting the results.

3. The word “pacemaker” is still used at some points (e.g. page 8, the first sentence). Please consider using the same term throughout the whole text.

Author Response

Manuscript ID: jcm-1699583

Title: Risk factors for mortality in cardiac implantable electronic device (CIED) infections: A systematic review and meta-analysis

Journal of Clinical Medicine

We thank the Editor for allowing us the opportunity to revise our manuscript and the Reviewer for the important and constructive comments. We have amended our paper in order to address the points raised by the Reviewers.

In the sections below, each of the points raised is identified and addressed with changes in the revised manuscript.

Reviewer #2:

Following a major revision the manuscript of Ngiam et al. has improved greatly. The authors have responded to the issues raised by the reviewer and have made the necessary changes in the text. However there are still some points that need to be considered.

  1. The references to the studies in the table should be numbered in Arabic numbers and appear as cited in the reference list. Listing them at the bottom of the table is redundant and just makes it harder to follow the text.

We thank the Reviewer for pointing this out. Indeed, we have now added the reference to the studies in the table in Arabic numerals, and these have been added to the main reference list.

  1. I do understand the value of logOR in this case but the authors should specify that a log OR has been used in the legends. I still think that this way of representing brings some unclarity and confusion as in the abstract and in the text the ORs are reported as linear measures. Perhaps this should be specified somehow in the text as well – e.g. by pointing that specifically in the statistical analysis section. It should also be specifically pointed out that the cut-off for LogOR is 0 as the reader gets confused very easily when interpreting the results.

We thank the Reviewer for this suggestion. Log OR is now reflected clearly in the legends of each figure and statistical analysis section as well for greater clarity. We have also specified that the cut-off for log OR is 0, and the dotted line at 0 in each of the figures also adds to that clarity.

  1. The word “pacemaker” is still used at some points (e.g. page 8, the first sentence). Please consider using the same term throughout the whole text.

We thank the Reviewer for this comment. We have used the appropriate term (anti-bradycardic CIED) instead of pacemaker throughout the text.

We thank the Reviewer and Editor for the kind and helpful comments. We hope the paper is now suitable for publication in the Journal.

Thank you

Dr Nicholas Ngiam, Dr William Kong